# Optimization of Exopolysaccharide Produced by *Lactobacillus plantarum* R301 and Its Antioxidant and Anti-Inflammatory Activities

**DOI:** 10.3390/foods12132481

**Published:** 2023-06-25

**Authors:** Junyong Wang, Jing Zhang, Henan Guo, Qiang Cheng, Zaheer Abbas, Yucui Tong, Tiantian Yang, Yichen Zhou, Haosen Zhang, Xubiao Wei, Dayong Si, Rijun Zhang

**Affiliations:** 1State Key Laboratory of Animal Nutrition, College of Animal Science and Technology, China Agricultural University, Beijing 100193, China; wangjy9722@cau.edu.cn (J.W.); zhangjing123@cau.edu.cn (J.Z.); chengqiangcool@163.com (Q.C.); zaheerabbas@cau.edu.cn (Z.A.); 15956910334@163.com (Y.T.); ytt2020@cau.edu.cn (T.Y.); redhoh@163.com (Y.Z.); s20223040758@cau.edu.cn (H.Z.); weixubiao01@126.com (X.W.); dayong@cau.edu.cn (D.S.); 2Department of Basic Medical Sciences, School of Medicine, Tsinghua University, Beijing 100084, China; ghn_657@163.com

**Keywords:** *Lactobacillus plantarum*, exopolysaccharide, response surface methodology, antioxidant, anti-inflammatory, postbiotics

## Abstract

In this study, the yield of exopolysaccharide (EPS) from *Lactobacillus plantarum* R301 was optimized using a single-factor experiment and response surface methodology (RSM). After optimization, the EPS yield was increased with a fold-change of 0.85. The significant factors affecting EPS production, as determined through a Plackett–Burman design and Central Composite Design (CCD), were MgSO_4_ concentration, initial pH, and inoculation size. The maximum yield was 97.85 mg/mL under the condition of 0.01% MgSO_4_, an initial pH 7.4, and 6.4% of the inoculation size. In addition, the EPS exhibited strong antioxidant activity, as demonstrated by its ability to scavenge DPPH, ABTS, and hydroxyl radicals. The scavenging rate was up to 100% at concentrations of 4 mg/mL, 1 mg/mL, and 2 mg/mL, respectively. Moreover, the EPS also exhibited reducing power, which was about 30% that of ascorbic acid when both tended to be stable with the increased concentration. These results suggest that *L. plantarum* R301 EPS possesses different antioxidant mechanisms and warrants further investigation. In addition to its antioxidant activity, the EPS also demonstrated good anti-inflammatory activity by inhibiting the inflammation induced by lipopolysaccharide (LPS) in RAW 264.7 cells, which could decrease nitric oxide (NO) production and expression of the proinflammatory cytokine *Il-6*. These findings suggest that *L. plantarum* R301 EPS could be used as a potential multifunctional food additive in the food industry.

## 1. Introduction

As a probiotic, lactic acid bacteria (LAB) are extensively found in nature [1,2]. People in many regions consume foods containing LAB [3], and LAB have been proven to have important roles in human health. However, in recent years, postbiotics have been widely studied since they are safer and easier to commercialize than probiotics [4,5]. Postbiotics have many functions, such as enhancing the epithelial barrier function [6,7], favorable modulation of the gut microbiota [8], immune responses [9,10], and systemic metabolism [11]. Moreover, studies on postbiotics focus on cell-free supernatants, exopolysaccharides (EPS), enzymes, cell wall fragments, and bacterial lysates.

EPS is a long-chain, high molecular weight polysaccharide with diverse structures and physicochemical properties [12]. These polysaccharides are secondary metabolites produced by different microorganisms, such as Lactobacillus and Bacillus, and are secreted to the extracellular space [13,14]. EPS has many advantages when compared with other postbiotics: a. cost-effectiveness, on the one hand, the medium required to produce EPS is cheap, and on the other hand, the extraction process of EPS is not complicated because it is secreted to the extracellular space; b. the abundance of species, the structure of EPS depends on the kinds of microorganisms and the composition of the culture medium [15]. In theory, a strain in different culture media could produce EPSs with different structures. This provides a quantitative basis for screening EPS with advantageous functions. For example, the hydrophobic group attached to the hydrophilic chain of EPS makes it amphiphilic and able to act as a natural emulsifier. The addition of EPS to low-fat mayonnaise could improve the rheological properties and stability of the product [16], and the emulsification activity of some EPSs was comparable to that of xanthan gum [17]. In addition, EPS could be used as a potential antioxidant in food processing because of its excellent antioxidant capacity [18]. Therefore, these properties of EPS have great potential to be used in food additives and functional food raw materials.

However, there are two problems before we further exploit EPS. The first is the yield of EPS. EPS production is influenced by numerous factors, such as the species of the microorganisms, the medium components (e.g., carbon source, nitrogen source), and different nutrients preferred by different microorganisms for EPS production [13,19]. In addition, the culture conditions (e.g., temperature, pH) also influence the production of EPS [20,21]. Moreover, the optimum conditions for EPS production are not always consistent with the growth of microorganisms. Even some EPS syntheses are performed by microorganisms only under stress conditions [22]. Whereas the optimal temperature for *Lactobacillus paracasei* growth is 37 °C, the temperature suitable for EPS production is 20 °C [20]. Therefore, systematic optimization of these factors is crucial to maximizing the EPS yield. The other problem is the biological activity of the EPS. The yield of EPS determines whether it is industrially viable, while the function determines the direction of research and application. Due to the diversity of EPS structures, the functions of EPS are also extensive, such as antioxidant, anti-inflammatory, emulsification activity, antimicrobial, anti-tumor, immunomodulatory, and cholesterol-lowering activities [23,24]. Therefore, uncovering the potential biological activities of EPS is crucial for further study and its application in the food industry.

This study also addresses both of these problems. The first part was to optimize the level of EPS production by *L. plantarum* R301, which was preserved in our laboratory. Single-factor optimization and response surface optimization were used for optimizing media composition and culture conditions to obtain the highest EPS yield while investigating the effect of different nutrients and culture conditions on EPS production. The other part is to investigate the potential biological activity of EPS for its subsequent research and application. Specifically, this work reported the antioxidant activity of the EPS in vitro, including DPPH, ABTS, hydroxyl radical-scavenging capacity, and reducing power. Moreover, the anti-inflammatory activity of EPS was investigated using an LPS-induced RAW 264.7 inflammation model.

## 2. Materials and Methods

### 2.1. Microorganism

The strain *L. plantarum* R301 was isolated from a traditional Chinese pickle sample from Xinjiang province, China. The strain was identified based on 16S rRNA sequencing and morphological characteristics. The strain was preserved at −80 °C with an MRS medium containing 50% glycerol.

### 2.2. Production and Separation of EPS

The MRS medium includes glucose (20 g/L), peptone (10 g/L), yeast extract (5 g/L), beef extracts (10 g/L), anhydrous sodium acetate (5 g/L), K_2_HPO4 (4 g/L), triamine citrate (2 g/L), Tween-80 (1 mL/L), MgSO_4_ (0.58 g/L), and MnSO_4_ (0.28 g/L).

The purification method of EPS was based on the procedure described by Adesulu-Dahunsi [3] with some modifications: to prepare EPS for analysis, the culture medium was first heat-treated at 100 °C for 10 min to eliminate bacteria and enzymes that could break down EPS. The medium was then clarified through centrifugation at 13,523 RCF for 20 min, and the resulting supernatant was treated with 5% (*w*/*v*) trichloroacetic acid at 4 °C for 6 h to remove any remaining protein. The denatured proteins were removed by performing another round of centrifugation at 13,523 RCF for 20 min. The resulting supernatant was mixed with three volumes of absolute ethanol and kept at 4 °C overnight to precipitate the crude EPS. Finally, the EPS was isolated through centrifugation at 13,523 RCF for 20 min. The crude EPS was dialyzed at 4 °C for 24 h using distilled water to remove any remaining components of the medium and other substances. The dialysis water was replaced every 8 h. After dialysis, the EPS solution was lyophilized to obtain the EPS powder and stored at −20 °C for subsequent studies.

The yield of crude EPS was determined using the phenol–sulfate acid method with glucose as a standard [25]. To perform the assay, 1 mL of sample was mixed with 1 mL of 5% phenol solution (*w*/*v*), followed by the addition of 5 mL of 98% sulfuric acid. The mixture was stirred and incubated at room temperature for 20 min, and the absorbance was measured at 490 nm. The yield of crude EPS was calculated based on the obtained absorbance value using a glucose standard curve.

### 2.3. Optimization of EPS Production

#### 2.3.1. Single-Factor Experiment Method

The highest yield of crude EPS was preliminary screened via single-factor analysis. This included carbon sources (glucose, lactose, fructose, sucrose), carbon source concentrations (2%, 3%, 4%, 6%, 8%, 10%), nitrogen sources (small peptide, beef extract, tryptone, whey protein powder), nitrogen source concentrations (1%, 2%, 3%, 4%), MgSO_4_ concentrations (0, 0.02%, 0.04%, 0.06%, 0.1%, 0.2%), MnSO_4_ concentrations (0, 0.02%, 0.04%, 0.06%, 0.1%, 0.2%), inoculation sizes (1%, 2%, 3%, 4%, 6%, 8%), fermentation temperatures (22, 27, 32, 37, 42 °C), and initial pHs (5.5, 6.0, 6.5, 7.0, 7.5, 8.0). In addition, the fermentation time was determined through the bacterial growth curve. The preliminary screening results would serve as a basis for the RSM experiment.

#### 2.3.2. Plackett–Burman Design

The dominant factors influencing crude EPS production were screened from the glucose concentration (A), small peptide concentration (B), MgSO_4_ concentration (C), MnSO_4_ concentration (D), initial pH (E), temperature (F), and inoculation size (G) based on the findings of the single-factor experiment. Each factor had two levels, high and low. The specific design method is shown in Table 1.

#### 2.3.3. Center Composite Design

The three dominant factors (MgSO_4_ concentration (C), initial pH (E), and inoculation size (G)) were further investigated using the Center Composite design (CCD) based on the results of the Plackett–Burman experiment. Each factor had three levels. The detailed design method is shown in Table 2. The experimental data were processed to obtain multivariate fitting equations and 3D graphic plots using Design-Expert 8.0.6 software (Stat-Ease, Minneapolis, MN, USA).

### 2.4. Antioxidant Activity Analysis

#### 2.4.1. DPPH Radical-Scavenging Activity

A modified method was used to determine the DPPH radical-scavenging activity [19]. Briefly, different concentrations (0.5, 1.0, 2.0, 4.0, 6.0, 8.0 mg/mL) of EPS were mixed with 2 mL of 0.2 mM DPPH solution (dissolved in absolute methanol). The mixture was incubated in the dark at room temperature for 30 min. The absorbance of the mixture was measured at 517 nm. The blank group was prepared by replacing DPPH with an equal volume of absolute methanol. The control group was prepared by replacing EPS with an equal volume of distilled water, and ascorbic acid was used as a positive control. The DPPH radical-scavenging activity was determined by using the following formula:$$\mathrm{Scavenging}\;\mathrm{activity}\;\left(\%\right)=\frac{A_{0}-\left(A_{1}-A\right)}{A_{0}}\times 100\%$$

*A* was the absorbance of the blank group, *A*_0_ was the absorbance of the control group, and *A*_1_ was the absorbance of the EPS group.

#### 2.4.2. ABTS Radical-Scavenging Activity

The method for measuring ABTS radical-scavenging activity was based on the procedure described by Wang [21] with some modifications. To generate the ABTS radical, solutions of 7 mM ABTS and 2.4 mM potassium persulfate were mixed in equal volumes and allowed to stand in the dark at room temperature for 16 h. Next, 500 μL of EPS of various concentrations (0.5, 1.0, 2.0, 4.0, 6.0, 8.0 mg/mL) was mixed with 1 mL of the ABTS radical solution and incubated for 10 min at room temperature in the dark. The absorbance of the resulting solution was measured at 734 nm. In the blank group, ABTS was replaced with an equal volume of distilled water, while in the control group, EPS was replaced with an equal volume of distilled water. Ascorbic acid was used as a positive control. The ABTS radical-scavenging activity was determined using the following equation:$$\mathrm{Scavenging}\;\mathrm{activity}\;\left(\%\right)=\frac{A_{0}-\left(A_{1}-A\right)}{A_{0}}\times 100\%$$

*A* was the absorbance of the blank group, *A*_0_ was the absorbance of the control group, and *A*_1_ was the absorbance of the EPS group.

#### 2.4.3. Hydroxyl Radical-Scavenging Activity

The hydroxyl radical-scavenging activity assay was conducted with some modifications based on the method described by Wang [21]. Briefly, EPS of various concentrations (0.5, 1.0, 2.0, 4.0, 6.0, 8.0 mg/mL) was mixed with 1.5 mL of 1.8 mM salicylic acid, 2 mL of 1.8 mM FeSO_4_, and 1 mL of 6 mM H_2_O_2_. After incubating the mixture for 30 min at 37 °C, the supernatant was collected via centrifugation at 845 RCF for 5 min, and its absorbance was measured at 510 nm. The blank group was prepared by replacing salicylic acid with an equal volume of distilled water, while the control group was prepared by replacing EPS with an equal volume of distilled water. Ascorbic acid was used as a positive control. The hydroxyl radical-scavenging activity was determined using the following equation:$$\mathrm{Scavenging}\;\mathrm{activity}\;\left(\%\right)=\frac{A_{0}-\left(A_{1}-A\right)}{A_{0}}\times 100\%$$

*A* was the absorbance of the blank group, *A*_0_ was the absorbance of the control group, and *A*_1_ was the absorbance of the EPS group.

#### 2.4.4. Reducing Power Analysis

The reducing power assay was conducted following a modified version of the method described by Liu [26]. Briefly, various concentrations of EPS (0.5, 1.0, 2.0, 4.0, 6.0, 8.0 mg/mL) were mixed with 0.2 M phosphate buffer (pH = 6.6) and 1% potassium ferricyanide (*w*/*v*), and the mixture was incubated at 50 °C for 20 min. After the addition of 10% trichloroacetic acid (*v*/*w*) and centrifugation at 845 RCF for 10 min, the supernatant was mixed with distilled water and 0.1% FeCl_3_ (*w*/*v*) and then incubated at room temperature for 10 min. The absorbance was measured at 700 nm, and ascorbic acid was used as a positive control.

### 2.5. Anti-Inflammatory Activity Analysis

#### 2.5.1. Cell Culture

The RAW 264.7 cell line was acquired from the American Type Culture Collection. The cells were sustained in Dulbecco’s modified Eagle’s medium (DMEM) supplemented with 10% fetal bovine serum (FBS) and 1% penicillin-streptomycin (HyClone, Logan, UT, USA) at 37 °C in a 5% CO_2_ humidified incubator. The culture medium was changed daily to ensure optimal growth conditions for the cells.

#### 2.5.2. Cytotoxicity Analysis

The toxicity of EPS towards RAW 264.7 cells was measured using the CCK-8 kit. RAW 264.7 cells were seeded at a density of 3.0 × 10^5^ cells per well in 96-well plates and incubated for 12 h. Then, the cells were treated with different concentrations of EPS (0, 20, 40, 60, 80, 100, 500, and 1000 μg/mL) for an additional 24 h. After that, 10 μL of CCK-8 solution was added to each well according to the kit instructions, and the plate was incubated in the dark at 37 °C for 2 h. The absorbance of the reaction solution was measured at 450 nm to calculate cell viability using the following equation:$$\mathrm{Cell}\;\mathrm{viability}\;\left(\%\right)=\frac{A_{S}-A_{B}}{A_{C}-A_{B}}\times 100\%$$

*A_S_* is the absorbance of the well containing cells, CCK-8, and EPS; *A_B_* is the absorbance of the well containing medium and CCK-8, but without cells; and *A_C_* is the absorbance of the well containing cells and CCK-8, but without EPS.

#### 2.5.3. Lipopolysaccharide (LPS)-Induced Inflammatory Macrophage Model

The LPS-induced inflammatory macrophage model was conducted following a modified version of the method described by Kwon [27]. LPS from *E. coli* 0111:B4 was purchased from Sigma-Aldrich (Saint Louis, MO, USA). RAW 264.7 cells were plated at a density of 3.0 × 10^5^ cells per well in 96-well plates and incubated for 12 h. After that, different concentrations of EPS were added to each well and incubated for an additional 12 h. Next, LPS was added to each well at a final concentration of 1 μg/mL and incubated for another 24 h.

#### 2.5.4. Determination of Nitric Oxide Content

To create the LPS-induced inflammatory macrophage model, the same procedure described above was followed. Nitric oxide (NO) content was quantified using the Griess reagent according to the manufacturer’s instructions. Specifically, 50 μL of Griess A and 50 μL of Griess B were sequentially added to 50 μL of medium supernatant, and the absorbance of the resulting reaction mixture was measured at 540 nm.

#### 2.5.5. RNA Isolation and Quantitative Real-Time PCR

The LPS-induced inflammatory macrophage model was constructed as in the above method with modifications; RAW 264.7 cells were seeded into 6-well plates at a density of 2.0 × 10^6^ cells per well, and the subsequent steps were performed as previously described. For RNA isolation, total RNA was isolated using TRIzol reagent (Solarbio, Beijing, China). The RNA integrity of all samples was evaluated via agarose gel electrophoresis and with a NanoDrop. All cDNA was synthesized using the HiScript^®^ III 1st Strand cDNA Synthesis Kit (Vazyme, Nanjing, Jiangsu, China) according to the manufacturer’s instructions. qPCR was performed using SYBR Green qPCR Master Mix (Bimake, Houston, TX, USA). The primers used in this study are listed in Table 3. Two-step amplification was used in this study: 30 s at 95 °C for preincubation, 40 cycles of 5 s at 95 °C for denaturing, and 30 s at 60 °C for extension, and then 1 cycle of 5 s at 95 °C, 60 s at 60 °C, and 1 s at 95 °C for melting. The normalization of gene expression was performed by comparing the expression of the target gene to that of the housekeeping gene *β*-actin. The results were then reported as the fold increase in gene expression relative to the control sample.

### 2.6. Statistical Analysis

All the experiments were performed in triplicate to minimize deviation, and the experiments of cytotoxicity analysis and NO content measurement were performed in sextuplicate. All the data were shown as the mean ± standard deviation (SD) of the independent repeats in each experiment. One-way ANOVA with Dunnett’s multiple-comparisons test was used for the single-factor experiment in the optimization study, cytotoxicity analysis, NO content measurement, and gene-expression-level analysis. The Plackett–Burman design and CCD experiment were analyzed using Design-Expert 8.0.6 software. A *p*-value < 0.05 was considered a significant statistical significance.

## 3. Results

### 3.1. Optimization for EPS Production

#### 3.1.1. Signal-Factor Experiment

The results of the signal-factor experiment are shown in Figure 1. The results included two parts: medium components and fermentation conditions. In the part of medium components, five kinds of carbon sources were chosen, glucose was the optimum carbon source, and the optimum supplemental concentration was 4% (Figure 1A,B). Four kinds of nitrogen sources were chosen, the small peptide was the optimum nitrogen source, and the optimum supplemental concentration was 2% (Figure 1C,D). The optimum MgSO_4_ and MnSO_4_ concentrations were 0.06% and 0.02% (Figure 1E,F), respectively. In the part of fermentation conditions, the optimum inoculum size was 3% (Figure 1G), the optimum medium initial pH was 7 (Figure 1H), and the optimum fermentation temperature was 27 °C (Figure 1I). In addition, the optimum fermentation time was determined based on the growth curve of *L. plantarum* R301. As shown in Figure 1G, EPS production increased rapidly from 0 to 9 h, leveled off after 9 h, and decreased slightly after 24 h. Therefore, we chose 24 h as the optimum fermentation time.

#### 3.1.2. Plackett–Burman Results

The results of the experimental design and significance analysis of the seven factors are shown in Table 4 and Table 5, respectively; *p* < 0.05 indicated a significant difference. A total of fifteen groups were designed in this experiment, which included three groups of center points. Two of the factors (initial pH and inoculation size) had a significant effect on the EPS yield, and both of the factors were positively correlated with EPS production. In addition, the MgSO_4_ concentration was the factor with the third lowest *p*-value (0.2576), and this factor was negatively correlated with EPS production. For the CCD experimental design, we chose the two significant factors (initial pH and inoculation size) and the lowest *p*-value in the remaining factors (MgSO_4_ concentration). Except for these three factors, the optimum conditions of the remaining factors were determined according to the single-factor results: glucose (3%), small peptide (2%), MnSO_4_ (0.06%), fermentation temperature (27 °C), fermentation time (24 h).

#### 3.1.3. Center Composite Design Results

According to the Plackett–Burman experimental analysis, we chose three factors (MgSO_4_ concentration, initial pH, and inoculation size) for further CCD experiments to evaluate the optimum conditions for the EPS yield. The results are shown in Table 6 and Table 7. A total of nineteen groups were designed in this experiment, which included five groups of center points. Table 6 shows the experimental results. This included the actual EPS yield obtained from the experiment and the theoretical prediction value obtained from the multiple regression equation.
(1)$$Y=-1412.15+13035.07\;X_{1}+354.50\;X_{2}+39.31\;X_{3}-828.11\;X_{1}X_{2}+15.34\;X_{1}X_{3}-2.50\;X_{2}X_{3}-351330\;X_{1}{}^{2}-22.33\;X_{2}{}^{2}-1.44\;X_{3}{}^{2}$$

The theoretical prediction and the actual value exhibited a high level of agreement, with an R-square value of 0.9288, indicating that the regression equation provides a good fit and effectively simulated the EPS production process of *L. plantarum* R301 (Table 7). As Table 6 shows, the highest EPS yield was in run 15 (97.68 mg/L), and the lowest one was in run 1 (61.24 mg/L). The yield difference between them was 36.44 mg/L. In Table 7, the model’s *p*-value was 0.0004, which meant that the model had a high degree of significance. The *p*-values of C, A^2^, B^2^, and C^2^ were all less than 0.01, indicating that the MgSO_4_ concentration, initial pH, and inoculation size exerted a considerable effect on the EPS yield, and the inoculation size had the most significant effect. However, the interaction among the three factors was not significantly different (*P_AB_* = 0.2197, *P_AC_* = 0.9243, and *P_BC_* = 0.1457). Moreover, we could observe the interaction effects through the 3D response surface plots and contour plots of the two factors in Figure 2. In addition, the non-significant lack of fit (*p* = 0.0993 > 0.05) suggested that the experimental processes were minimally influenced by other unidentified factors.

Above all, the optimum conditions for producing EPS were determined according to the CCD experiment as follows: the MgSO_4_ concentration was 0.01%, initial pH was 7.4, and inoculation size was 6.4%. The predicted highest EPS yield was 97.85 mg/L.

### 3.2. Anti-Oxidation Activity of EPS In Vitro

#### 3.2.1. DPPH Radical-Scavenging Activity

The DPPH radical-scavenging activity of EPS is shown in Figure 3A. At concentrations ranging from 0 to 4 mg/mL, the scavenging capacity of ascorbic acid for free radicals was significantly greater than that of EPS. For instance, at a concentration of 0.5 mg/mL, the scavenging activity of ascorbic acid was measured to be 94.44% ± 0.10%, while the scavenging activity of EPS was only 29.27% ± 1.08%. In addition, the scavenging activity of EPS was positively correlated with its concentration range of 0–4 mg/mL. At a concentration of 4 mg/mL, the scavenging activity of EPS (97.69% ± 0.27%) was higher than that of ascorbic acid (94.74% ± 0.18%). At a concentration of 4 mg/mL, there was no significant difference in the scavenging activity between EPS and ascorbic acid.

#### 3.2.2. ABTS Radical-Scavenging Activity

The ABTS radical-scavenging activity of EPS is shown in Figure 3B. The scavenging activities of EPS and ascorbic acid were not much different in the given concentration range. EPS (96.26% ± 1.68%) was lower than ascorbic acid (100%) only at 0.5 mg/mL.

#### 3.2.3. Hydroxyl Radical-Scavenging Activity

The hydroxyl radical-scavenging activity of EPS is shown in Figure 3C. The hydroxyl radical-scavenging ability of ascorbic acid was not as strong as that with ABTS and DPPH radicals. The concentration to achieve 100% scavenging ability was 1 mg/mL, while that for ABTS was 0.125 mg/mL and that for DPPH was 0.25 mg/mL. The scavenging ability of EPS was 96.52% ± 1.20% at 2 mg/mL. At a concentration of 2 mg/mL, the scavenging ability increased slowly (2 mg/mL → 8 mg/mL, 96.52% ± 1.20% → 99.13% ± 0.49%).

#### 3.2.4. Reducing Power Analysis

The reducing power of EPS is shown in Figure 3D. EPS was much lower than ascorbic acid. Ascorbic acid increased slowly after 0.25 mg/mL, while EPS was after 2 mg/mL. The reducing power of EPS was about 30% that of ascorbic acid when both tended to be stable with the concentration increasing.

### 3.3. The Cytotoxicity of EPS

The cytotoxicity of EPS against RAW 264.7 cells was determined using the CCK-8 assay. Within the concentration range of 20–80 μg/mL, the viability of RAW 264.7 cells was not significantly affected by EPS (Figure 4). When the concentration increased, there was a gradual decline in the viability of RAW 264.7 cells. We chose the concentrations of 20, 40, and 60 μg/mL for the anti-inflammatory experiment according to this result.

### 3.4. Anti-Inflammatory Activity of EPS in LPS-Induced RAW 264.7 Cells

#### 3.4.1. Effect of EPS on NO Production

The result of NO production is shown in Figure 5A. LPS stimulation markedly elevated the levels of NO compared to those in the control group (*p* < 0.05), while this was significantly inhibited with EPS treatment compared with levels in the LPS group (*p* < 0.05). The inhibitory ability was enhanced with an increase in the EPS concentration, but this was not significant.

#### 3.4.2. Effect of EPS on Transcription Levels of *Inos*, *Nf-κb*, *Cox-2*, and *Il-6*

QPCR results of the above genes are shown in Figure 5B–E. Contrary to the results of the NO production assay, the expression level of inducible nitric oxide synthase (*Inos*) in the EPS treatment groups did not change significantly when compared with that in the LPS group (Figure 5B). LPS treatment significantly increased the expression level of nuclear factor κB (*Nf-κb*) compared with that in the control group (*p* < 0.05), but there was no difference between the EPS groups and the control group (Figure 5C). The expression level of cyclooxygenase-2 (*Cox-2*) in the EPS groups decreased when compared with that in the LPS group. However, the expression level and EPS concentration had a positive correlation among the three groups (Figure 5D). There was a 60-fold difference in interleukin 6 (*Il-6*) expression between the LPS treatment group and the control group. In the EPS groups, the expression of Il-6 was inhibited when compared with that in the LPS group, and there was no significant differences observed among the three EPS groups (Figure 5E).

## 4. Discussion

EPS production is affected by many factors, such as genetics, the culture medium, and fermentation conditions. The genetics factor mostly depends on the strain types, such as *Pseudoalteromonas agarivorans* (2783.6 mg/L) [28], *Lactobacillus rhamnosus* (210.28 mg/L) [29], and *L. plantarum* (280.105 mg/L) [30], while the culture medium and fermentation conditions need to be optimized for the specific strain. The present study optimized EPS production by *L. plantarum* R301 using a single-factor experiment and RSM. The single-factor experiment preliminarily obtained the conditions for EPS production: 4% glucose, 2% small peptide, 0.06% MnSO_4_, 0.02% MgSO_4_, fermentation temperature of 27 °C, 3% inoculum size, initial pH of 7.0, and the fermentation time of 24 h (Figure 1). The RSM experiment further optimized three factors (MgSO_4_ concentration, initial pH, and inoculation size) to obtain the maximum EPS yield (Table 4, Table 5, Table 6 and Table 7). The carbon and nitrogen sources were identified as the primary influential factors in EPS production, and different strains prefer different nutrient sources. This study shows that glucose is the most favorable carbon source for EPS production. However, choosing the optimum nitrogen source was not an easy job, because most of the nitrogen sources contained polysaccharides that could be detected via the phenol–sulfate method, which would contaminate the results of EPS yield optimization. The small peptide that contained a variety of short peptides was processed in our laboratory. It was used as the nitrogen source because it contained fewer polysaccharides and could better promote the growth of *L. plantarum* R301. The impact of metal ions on EPS production is mainly reflected by effects on the growth of bacteria. In this study, excessive or undersized MgSO_4_ was not suitable for the growth of *L. plantarum* R301, which directly led to the reduction of EPS production (Figure 1F). In addition, the MgSO_4_ concentration was the significant factor that affected EPS production in the RSM experiment, which means that it might have some other functions for EPS production. Some metal elements are important for bacterial growth and metabolism [31]. Whether the MgSO_4_ concentration promotes the production of EPS by affecting the metabolism of *L. plantarum* R301 needs further study. The optimum fermentation temperature of 27 °C was interesting because the optimum growth temperature of *L. plantarum* R301 was 37 °C. This was not the same as most of the relevant studies, which had the same temperature [21,29]. One theory is that microorganisms produce EPS to protect themselves under stress [32,33]. For this study, *L. plantarum* R301 growth restriction may be due to the low temperature as a stressor, thus improving EPS production to protect itself. The research of Bengoa et al. [20] and Magdalena et al. [22] also supported our hypothesis in that the optimum temperatures were 20 °C and 25 °C, respectively. Here, we optimized the conditions of EPS production in *L. plantarum* R301, and the EPS production increased from 53.34 mg/L to 97.85 mg/L, which increased by 84.70%.

As we discussed, finding EPS with good functional activity is the basis for further studies. In this research, we mainly studied the functions of EPS in two aspects: antioxidant and anti-inflammatory. Reactive oxygen species (ROS) have significant functions in various cellular processes, including apoptosis, cell signaling, genetic regulation, and the transportation of ions [34]. Although reactive oxygen species (ROS) have important physiological functions, an excess of ROS can cause various harmful effects on the body. These include damaging DNA, promoting the development of cancer, and accelerating cellular degeneration, which can lead to several diseases, such as inflammation, lung injury, and other disorders [35,36,37]. Certain antioxidants, such as butylated hydroxyanisole (BHT) and butylated hydroxytoluene (BHA) have been demonstrated to effectively scavenge excessive free radicals. However, it is important to note that these antioxidants may also have harmful effects on the body [34,38]. For this reason, it is necessary to find novel green and safe antioxidants. In this study, we assessed three kinds of free radicals: DPPH, ABTS, and ·OH (Figure 3A–C). The scavenging rate of EPS for these three free radicals could reach 100% at different concentrations. To some extent, the EPS of *L. plantarum* R301 had strong antioxidant activity, because most known EPS does not have such strong scavenging activity. For example, the EPS of *L. plantarum* KX041 could reach approximately 80% of the scavenging rate of the three free radicals at a concentration of 8 mg/mL [21]. At a concentration of 8 mg/mL, the EPS of *Bacillus velezensis* SN-1 exhibited a scavenging rate of approximately 60% for the three free radicals [13]. In addition, the EPS of *L. plantarum* R301 also had a certain reducing power (Figure 3D), which indicated that the EPS had different antioxidant mechanisms [39]. However, the free radical-scavenging activity can only be used for a preliminary evaluation. In vitro and in vivo experiments do not always produce the same effects [40]. For instance, antioxidant peptides derived from ovotransferrin, which were identified using the oxygen-radical absorbance capacity assay, were found to lack antioxidant activity in endothelial cells [41]. For this reason, more comprehensive studies on the antioxidant activity of EPS in vivo are needed in the future.

Lipopolysaccharide (LPS) is the most extensively studied type of pathogen-associated molecular pattern (PAMP). When present, LPS can bind to and activate toll-like receptor-4 (TLR-4), which is a type of pattern recognition receptor present on the surface of specific immune cells, such as macrophages, neutrophils, and dendritic cells. The binding of LPS to TLR-4 triggers a signaling pathway that activates multiple transcription factors and ultimately induces an inflammatory response. The immune response triggered by this process involves the synthesis of pro-inflammatory cytokines, which play a vital role in the immune system’s capacity to combat invading pathogens. This inflammatory response includes the production of pro-inflammatory cytokines, which play a crucial role in the immune system’s ability to eliminate invading pathogens [42]. However, excessive inflammation can disturb the immunomodulatory balance, such as the cytokine storms triggered by an overactive inflammatory response, which can further aggravate the disease and even cause death. Therefore, controlling inflammation is an effective way to prevent further deterioration of the disease. Here, we used the LPS-induced macrophage inflammation model to evaluate the anti-inflammatory activity of EPS. We found that the EPS could decrease production of the pro-inflammatory cytokine Il-6 (Figure 5E). In addition, the expression of Cox-2, which is known as a major mediator of inflammation, was reduced by EPS treatment (Figure 5D). These results suggested that the EPS had anti-inflammatory effects, but the exact mechanism was unclear. There are several underlying anti-inflammatory mechanisms of EPS, such as EPS being able to suppress the expression of TLR4 [27]. The previous study showed that EPS could reduce NO production (Figure 5A), which was one of the ROS in LPS-induced macrophages. It was easy to relate this result to the excellent antioxidant activity of EPS [43]. However, whether there is a relationship between antioxidant activity and the anti-inflammatory activity of EPS and how they interact need further study.

## 5. Conclusions

In the present study, we figured out the main medium components and culture conditions that affect EPS production by *L. plantarum* R301 and optimized them using RSM. The maximum EPS yield increased from 53.34 mg/L before optimization to 97.85 mg/L after optimization, which increased by 84.70%. Moreover, the EPS exhibited strong antioxidant activity with a high free radical-scavenging rate with DPPH, ABTS, and hydroxyl free radicals. Reducing power analysis showed that the antioxidant mechanisms of EPS were varied. In addition, the EPS could alleviate the LPS-induced macrophage inflammation, which indicated that the EPS has anti-inflammatory properties. Therefore, the EPS produced by LAB exerted anti-inflammatory and antioxidant effects, indicating that it has the potential to be used as a food additive or functional food ingredient in the food industry. Further investigation will concentrate on the purification of EPS, characterization of its structure, and an exploration of the underlying mechanisms of its anti-inflammatory and antioxidant activities.

## Figures and Tables

**Figure 1 foods-12-02481-f001:**
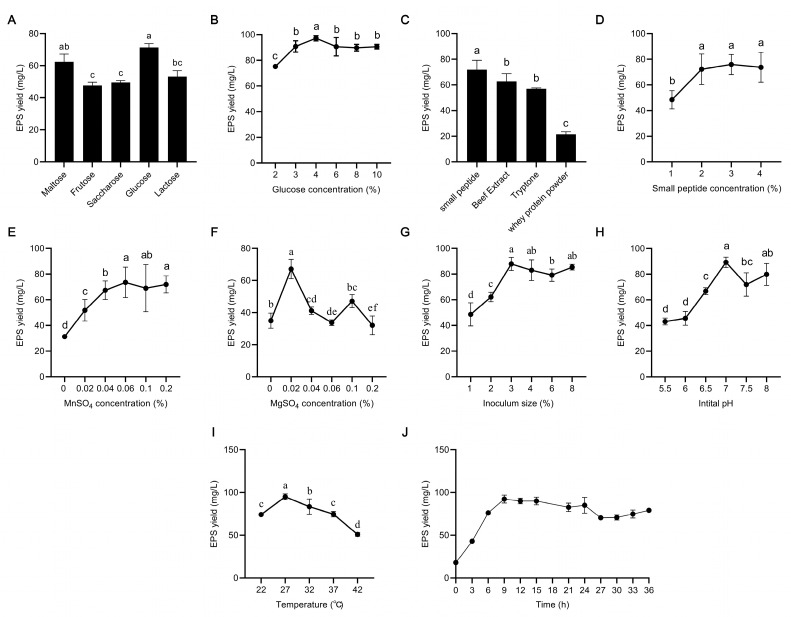
Results of single-factor optimization of EPS production by *L. plantarum* R301: (**A**) carbon sources, (**B**) carbon concentration, (**C**) nitrogen source, (**D**) nitrogen concentration, (**E**) MnSO_4_ concentration, (**F**) MgSO_4_ concentration, (**I**) fermentation temperature, (**G**) inoculum size, (**H**) initial pH of medium, and (**J**) fermentation time. Different lowercase letters indicate significant differences (*p* < 0.05).

**Figure 2 foods-12-02481-f002:**
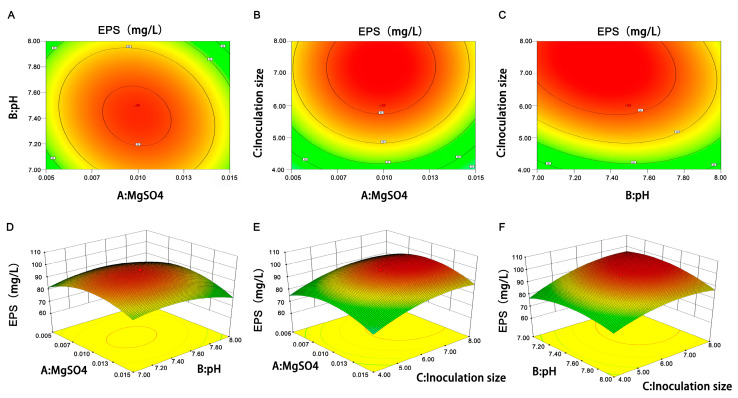
3D response surface plots and contour plots showing the interactive effects of the (**A**,**D**) MgSO_4_ concentration and initial pH of medium, (**B**,**E**) inoculum size and MgSO_4_ concentration, and (**C**,**F**) inoculum size and initial pH of medium.

**Figure 3 foods-12-02481-f003:**
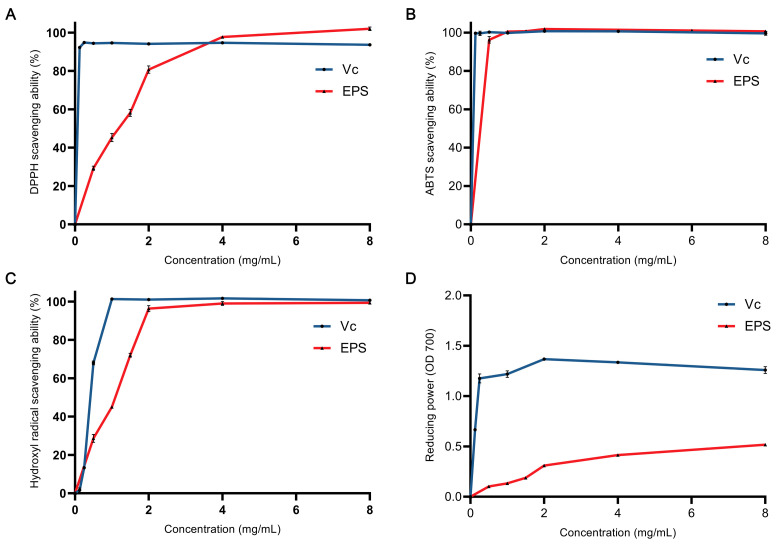
Antioxidant activity of EPS against the following: DPPH radical (**A**), ABTS radical (**B**), hydroxyl radical (**C**), and reducing power (**D**).

**Figure 4 foods-12-02481-f004:**
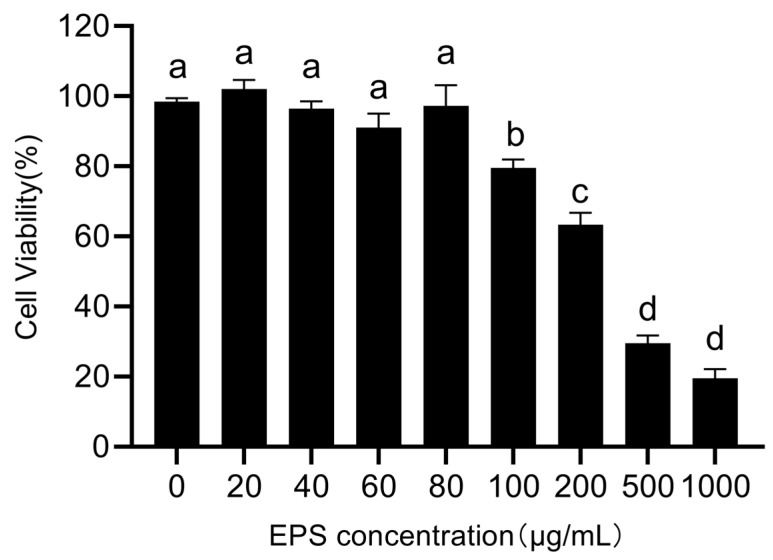
Effect of EPS on the viability of RAW 264.7 cells. Different lowercase letters indicate significant difference (*p* < 0.05).

**Figure 5 foods-12-02481-f005:**
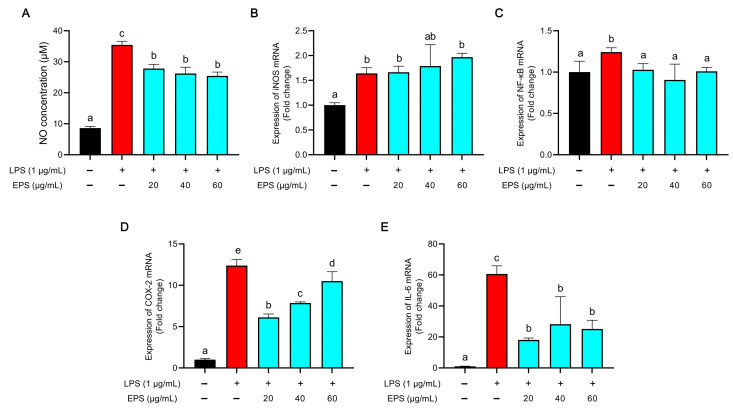
Anti-inflammatory effect of EPS: (**A**) NO production, (**B**–**E**) mRNA expression of *Inos* (**B**), *Nf-κb* (**C**), *Cox-2* (**D**), and *Il-6* (**E**) in EPS-treated RAW 264.7 cells. Different lowercase letters indicate significant difference (*p* < 0.05).

**Table 1 foods-12-02481-t001:** Factors and levels of Plackett–Burman design.

Factors	Number	Low Level (−)	High Level (+)
Glucose (%)	A	4	6
Small peptide (%)	B	1	3
MgSO_4_ (%)	C	0.02	0.06
MnSO_4_ (%)	D	0.04	0.08
pH	E	6	7
Temperature (°C)	F	27	37
Inoculation size (%)	G	3	7

**Table 2 foods-12-02481-t002:** Factors and levels of Center Composite design.

Factors	Number	Levels		
		−1	0	+1
MgSO_4_ (%)	X_1_	0.02	0.04	0.06
pH	X_2_	6	6.5	7
Inoculation size (%)	X_3_	3	5	7

**Table 3 foods-12-02481-t003:** Sequences of the primers used for RT-PCR assays.

Gene		Sequence (5′–3′)	Length
*Nf-κb*	F	GTCTTACACTTAGCCATCATCCACCTC	27
	R	ATCCTCTACTACATCTTCCTGCTTGGT	27
*Cox-2*	F	ATCAGGTCATTGGTGGAGAGGTGTAT	26
	R	TGCTGGTTTGGAATAGTTGCTCATCA	26
*Il-6*	F	TCTTGGGACTGATGCTGGTGA	21
	R	TTGGGAGTGGTATCCTCTGTGAA	23
*Inos*	F	TGGAGCGAGTTGTGGATTGTCCTA	24
	R	GCCTCTTGTCTTTGACCCAGTAGC	24
*β-actin*	F	TCACTATTGGCAACGAGCGGTTC	23
	R	CAGCACTGTGTTGGCATAGAGGTC	24

**Table 4 foods-12-02481-t004:** Plackett–Burman experimental design results.

Run	A	B	C	D	E	F	G	EPS (mg/L)
1	1	1	−1	1	1	1	−1	85.69
2	−1	1	1	−1	1	1	1	101.67
3	1	−1	1	1	−1	1	1	88.79
4	−1	1	−1	1	1	−1	1	106.39
5	−1	−1	1	−1	1	1	−1	56.88
6	−1	−1	−1	1	−1	1	1	79.82
7	1	−1	−1	−1	1	−1	1	103.74
8	1	1	−1	−1	−1	1	−1	41.07
9	1	1	1	−1	−1	−1	1	63.61
10	−1	1	1	1	−1	−1	−1	49.29
11	1	−1	1	1	1	−1	−1	46.70
12	−1	−1	−1	−1	−1	−1	−1	45.67
13	0	0	0	0	0	0	0	59.12
14	0	0	0	0	0	0	0	51.53
15	0	0	0	0	0	0	0	66.83

**Table 5 foods-12-02481-t005:** Significance analysis of factors influencing EPS production.

Source	Regression Analysis Coefficient	F Value	*p*-Value
Model		5.14	0.0232 *
A	−0.84	0.05	0.8284
B	2.18	0.34	0.5799
C	−4.62	1.52	0.2576
D	3.67	0.96	0.3601
E	11.07	8.72	0.0213 *
F	3.21	0.73	0.4201
G	18.23	23.65	0.0018 **

Note: R-square = 0.9370, * *p* < 0.05, ** *p* < 0.01.

**Table 6 foods-12-02481-t006:** Design and results of Central Composite Design.

Run	A	B	C	EPS (mg/mL)	
	MgSO_4_ (%)	pH	Inoculation Size (%)	Actual	Predicted
1	−1	−1	−1	61.24	67.03
2	1	−1	−1	70.93	69.76
3	−1	1	−1	69.21	72.42
4	1	1	−1	66.02	66.86
5	−1	−1	1	83.20	85.51
6	1	−1	1	88.91	88.85
7	−1	1	1	76.58	80.89
8	1	1	1	78.60	75.96
9	−1.68	0	0	79.89	72.12
10	1.68	0	0	66.94	70.26
11	0	−1.68	0	85.97	83.40
12	0	1.68	0	78.97	77.09
13	0	0	−1.68	71.79	68.16
14	0	0	1.68	92.16	91.34
15	0	0	0	97.68	96.03
16	0	0	0	95.88	96.03
17	0	0	0	94.67	96.03
18	0	0	0	95.29	96.03
19	0	0	0	95.88	96.03

**Table 7 foods-12-02481-t007:** ANOVA analysis of regression equation.

Factors	Sum of Squares	Degree of Freedom	Mean Square	F Value	*p*-Value
Model	2313.37	9	257.04	13.04	0.0004 **
A-MgSO_4_	4.16	1	4.16	0.21	0.6568
B-pH	48.12	1	48.12	2.44	0.1526
C-Inoculation size	648.76	1	648.76	32.92	0.0003 **
AB	34.29	1	34.29	1.74	0.2197
AC	0.19	1	0.19	0.01	0.9243
BC	49.99	1	49.99	2.54	0.1457
A^2^	1053.05	1	1053.05	53.43	0.0001 **
B^2^	425.45	1	425.45	21.59	0.0012 **
C^2^	452.47	1	452.47	22.96	0.0010 **
Residual	177.38	9	19.71		
Lack of fit	172.33	5	34.47	3.88	0.0993
Net errors	5.04	4	1.26		
Total deviation	2490.74	18			

Note: R-square = 0.9288, ** *p* < 0.01.

## Data Availability

No new data were created or analyzed in this study. Data sharing is not applicable to this article.

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
