# Peer review of "Optimization of Exopolysaccharide Produced by Lactobacillus plantarum R301 and Its Antioxidant and Anti-Inflammatory Activities"

_foods, 2023, doi:10.3390/foods12132481_

Round 1

Reviewer 1 Report

The manuscript describes optimization of medium composition for maximising the EPS production of L. plantarum R301, followed by determining antioxidant activity of the EPS in vitro and the anti-inflammatory activity of EPS. The novelty of the study can be highlighted in the introduction.

1.      In the result, EPS production leveled off after 9 hours, why authors chose 24 hours as the optimum fermentation time instead of 9 hours?

2.      Line 245, “As shown in Figure 1 G,”. the figure is wrong, please check.

3.      Although MgSO4 is the third lowest P-value (0.2576), it was insignificant. Why author insisted to choose MgSO4 in the optimization study? This led to the insignificant model term in ANOVA during optimization. Any justification?

4.      Need to state what is LPS in the methodology.

The quality of English language can be improved.

Reviewer 2 Report

The manuscript entitled, "Optimization of Exopolysaccharide produced by Lactobacillus plantarum R301 and its antioxidant and anti-inflammatory activities" decribes the optimized yield of EPS from Lactobacillus plantarum R301 and its bioactive potential, however, the manuscript needs to be critically revised before consideration.

Introduction: Background information is not sufficient, authors should highlight different methods on extraction and comparison of reported yield among them.

Methods: please change RPM into g force or RCF. 

Please provide a reference for the isolation of EPS in method section. 

How EPS was converted into powdered form for the preparation of different concentrations. 

The formula for DPPH assay is not valid please recheck and define each parameter, as it should control-sample/control*100.

similarly, please check for other antioxidant assays.

Provide the reference for all the methods used.

Provide the reference for each primer used for PCR.

Any other method used for EPS quantification or study about its chemical nature , FTIR, HPLC or LCMS would add good information on it. 

What is reason for antioxidant potential of EPS?? 

Please clearly mention the novelty statement in abstract, introduction and conclusion and justify the application in food science. 

The minor language editing can improve the written expression of the manuscript. 
